# Load Frequency Control Assessment of a PSO-PID Controller for a Standalone Multi-Source Power System

**Boopathi Dhanasekaran** [1,*], **Jagatheesan Kaliannan** [1], **Anand Baskaran** [2], **Nilanjan Dey** [3] and **João Manuel R. S. Tavares** [4]

1   Paavai Engineering College, Namakkal 637 018, India
2   Hindusthan College of Engineering and Technology, Coimbatore 641 032, India
3   Techno International New Town, Kolkata 700 156, India
4   Instituto de Ciência e Inovação em Engenharia Mecânica e Engenharia Industrial, Departamento de Engenharia Mecânica, Faculdade de Engenharia, Universidade do Porto, 4200-465 Porto, Portugal
*   Correspondence: boopathime@gmail.com

**Abstract:** The performance of load frequency control (LFC) for isolated multiple sources of electric power-generating units with a proportional integral derivative (PID) controller is presented. A thermal, hydro, and gas power-generating unit are integrated into the studied system. The PID controller is proposed as a subordinate controller to stabilize system performance when there is a sudden demand on the power system. The particle swarm optimization (PSO) algorithm is used to obtain optimal gain values of the proposed PID controller. Various cost functions, mainly integral time absolute error (ITAE), integral absolute error (IAE), integral squared error (ISE), and integral time squared error (ITSE) were used to optimize controller gain parameters. Furthermore, the enhancement of the PSO technique is proven by the performance comparison of conventional, differential evolution (DE) algorithm- and genetic algorithm (GA)-based PID controllers for the same system. The results show the PSO-PID controller delivers a faster settled response and the percentage improvement of the proposed technique over the conventional method is 79%, over GA is 55%, and over DE is 24% in an emergency in a power system.

**Keywords:** differential evolution algorithm; genetic algorithm; particle swarm optimization; integral time absolute error; PID controller; cost function; load frequency control

## 1. Introduction

The electric power system consists of both power production and distribution, which is the power generation on the users load demand. As a result of globalization and technological advancement, the demand for electricity from customers is accumulating daily. To fulfill load demand, the generating capacity is increased by constructing new power plants and upgrading existing ones. When implementing a sophisticated power network, the power grid has several issues, such as voltage and frequency deviation. Consumers use electricity in a nonlinear manner. As a result, the power production varies proportionally with load demand to ensure system performance stability. When a system or any interconnected system has a rapid increase in power demand, it impacts the stability of the whole power-generating unit.

The frequency is a crucial factor in the quality of the power system, and the LFC method addresses the issue of frequency variation. To execute the LFC scheme, a secondary controller must be included in the system to achieve greater performance and recover the given power supply [1,2]. The controller gain must be optimized due to the inability of the secondary controllers to reach the desired result. In this study, various optimization strategies were designed and used by the literature research to improve the controller gain settings. In the following, the response of secondary controllers and optimization strategies that have been implemented and explored are discussed.

### 1.1. Literature Review

To solve the LFC crisis of interconnected thermal power systems, [3] proposed a PSO-PID controller. The superiority of the projected controller was evident when its performance was compared to the performance of the hill climbing (HC)- and genetic algorithm (GA)-tuned controllers. The author [4], used the ant colony optimization (ACO) technique with the PID controller for a standalone nuclear power plant LFC and the response was compared using the trial and error method. The GA–PID controller was applied [5] for the LFC of an isolated thermal power plant. The superiority of the PID controller performance was exposed relative to the proportional integral (PI) controller. The AGC of the grid-connected power network was investigated by applying the hybrid fractional order fuzzy intelligent controller (FOFP-PID). Additionally, its supremacy was examined by comparing its performance with fractional order PID (FOPID) [6]. The PID controller was tuned using the search group algorithm (SGA) to overcome the AGC emergency of an interconnected power system in [7].

A hybrid GA (hGA) with the PSO technique was utilized to improve the AGC of the grid-connected system during an emergency loading in the power-generating unit [8]. Load frequency control of multi-source power system is examined by applying differential evolution algorithm tuned parameters based controller [9]. A PI Controller is designed for single area power system frequency control, gain value of controller is tuned by Stochastic Particle Swarm Optimization [10]. The authors of [11] developed a super twisting sliding mode controller (ST-SMC) for an LFC interconnected thermal power system to boost system efficiency during sudden load demand situations. The authors of [12] developed a hybrid many optimizing liaisons–gravitational search algorithm (hMOL–GSA)-based fuzzy PID (FPID) which was studied for the AGC of an interlinked thermal power plant. The ACO–PID controller was proposed for a single-area non-reheated thermal power plant by [13] to improve the system performance, [14] developed a hybrid fuzzy PID (hFPID) controller that was for the LFC crisis of a grid-connected power network. The authors of [15] applied a bacterial foraging (BF)-optimized fractional order fuzzy PID (FOFPID) controller for the LFC of several interconnected sources for electricity and its performance was compared with conventional and FPID controllers. The authors of [16] proposed the moth flame optimization (MFO)–proportional integral double derivative (PIDD) controller for rectifying the AGC of a grid-connected power network that included thermal, hydro, and nuclear power units.

The PSO technique associated with the BF-optimized PID controller was designed for solving the LFC for interconnected power networks, which included thermal and PV [17]. The PSO-associated multi-agent reinforcement learning (MARL) approach was implemented for the LFC of a grid-connected power network [18]. The authors of [19] designed the grasshopper optimization algorithm (GOA)-tuned fuzzy proportional derivative–fuzzy proportional integral (FPD–FPI) controller which was proposed for the AGC of a grid-connected power-generating unit consisting of a thermal power-generating unit with renewable energy resources, mainly wind and solar power units. The authors of [20] developed the adaptive artificial neural network (ANN)-based PID controller that was designed for the LFC of a thermal unit which was incorporated with different DG includes wind turbine generators (WTGs), battery energy storage system (BESS), aqua electrolyzer (AE), diesel engine generators (DEGs), and fuel cells (FCs).

A chaos-based firefly algorithm-regulated PID controller was applied for the LFC of an interlinked power network [21]. The authors of [22] developed the lion algorithm (LA) to optimize the controller gain of the FOPI controller for resolving LFC issues in a grid-connected power network. An I-PD controller was proposed to overcome the AGC problem of a three-area power grid, with the gain of the controller found using the fitness dependent optimizer (FDO) technique [23]. A novel technique, called Harris–Hawks optimization (HHO), was developed by [24] for a tilt integral derivative + filter (TID + F) controller and it was applied for an interlinked power network LFC with many DC power-generating units. [25], considered the shuffled frog-leaping algorithm (SFLA) for the LFC in a two-area

interlinked power network with a PID controller. A Hybrid PSO algorithm is utilized for in-depth analysis of the energy deduction test statistic in radio network [26].

The researcher in [27] implemented the coordinative optimization technique in a microgrid for minimizing the operation cost for scheduling the behavior of components in the microgrid and its performance was compared with traditional approaches, mainly the day-ahead economical dispatch strategy and heuristic logic algorithm, which proved the supremacy of the proposed technique. A hybrid algorithm was suggested by [28] for solving distributed storage allocation problems for minimizing storage costs. By applying the proposed technique, the cost of network resources and storage were effectively minimized. The authors of [29] suggested a machine learning algorithm (MLA) for analyzing the data collected by using the Internet of Things used in healthcare systems and smart cities. The major need for analyzing the data was to derive useful inferences from the analysis. K means algorithms were applied for obtaining a superior clustering performance by [30], and the performance of the proposed technique was confirmed by conducting various test analyses. Finally, the result analysis shows that it yields lesser execution time with superior clustering fitness and lesser sum of squared errors (SSE) over other algorithm techniques. The authors of [31] applied the flower pollination algorithm (FPA) to optimize the PID controller gain parameters of LFC issues in the interlinked power grid and the result was compared to the ones of the GA–PID and PSO–PID controllers to confirm the supremacy of the proposed FPA technique. The firefly algorithm was used by [32] to perform the AGC in a five-area power system and the performance was analyzed for the firefly algorithm (FFA)–PID controller against the responses with the GA–PID and PSO–PID controllers. The authors of [33] discussed the PSO-optimized PID regulator for a single area power grid LFC problem. Using a PSO optimization technique with four distinct cost functions, a PID controller was built for the LFC of an isolated power network. The authors of [34] explored a freestanding multiple source power system with the aid of a PSO–PID controller, and compared the results with the ones of a conventional technique. The author designed and developed a hybrid PSO–GSA algorithm-tuned PID controller for frequency regulation of an independent microgrid system. Additionally, the behavior of the proposed method was evaluated by comparing the response with PSO-tuned controller performance [35]. A fractional order controller was designed and applied in shipboard microgrids for the regulation of system frequency and gain values of compellers were tuned by utilizing a direct search algorithm. Further, effectiveness was evaluated by applying parameter variations and a time delay [36]. In [37] a fractional order fuzzy controller was developed and implemented in hybrid power systems (renewable power). The performance was evaluated by comparing the response with the PID controller and fractional order PID controller. Frequency improvement and regulation of a dual-area interconnected power system was studied by considering a PSO-tuned fuzzy FOPI–FOPD controller in [38]. The authors designed a movable damped wave algorithm-tuned FOPID controller and applied it in an interconnected multi-area power system for load frequency regulation of the system. In this work, renewable energy resources were also considered [39]. A hybrid microgrid power generating system frequency controller was analyzed by considering an optimal fuzzy PIDF controller in [40].

The literature review successfully suggests that the power system performance oscillates and is affected due to unexpected load demands in the power-generating unit. Due to this, LFC/AGC issues can occur in the power system. These issues are overcome by applying many optimization techniques in different controllers to optimize their gain under different situations and criterion [3–34]. A summary of the literature review is tabulated in Table 1.

**Table 1.** Summary of the literature review.

| Optimization Technique/Secondary Controller | Source for System | Study | Ref. |
|---|---|---|---|
| PSO–PID | Thermal | Results compared with GA and hill climbing | [3] |
| GA–PID | Thermal | Performance of PI and PID with/without GRC | [5] |
| Search group algorithm–PID | Thermal–hydro–gas | FA–PID results compared | [7] |
| PSO–PID | Thermal–hydro | GA–PID results compared with PSO | [8] |
| DE–PID | Thermal–hydro–gas | I, PI, and PID controller performance analyzed | [9] |
| Stochastic PSO–PI | Thermal | IAE, ISE, and ITAE cost functions performance analyzed | [10] |
| Hybrid bacteria foraging optimization algorithm–PID | Thermal–PV | PSO and BFO result were studied | [17] |
| Artificial neural network (ANN)–PID | Distributed generation sources (WTG, DEG, AE, FC) | Grasshopper optimization algorithm was utilized for supremacy of the proposed technique | [20] |
| PSO–PID | Thermal–hydro–gas–nuclear–PV | Conventional PID controller results compared with PSO–PID | [33] |
| PSO–PID | Thermal–PV–wind | Conventional I, PI, and PID controller performance compared | [34] |
| Stochastic PSO–PID | Thermal | Reliability of the technique verified by changing system parameters | [41] |

The above table provides a description of the literature related to the proposed work and it shows that numerous professionals use the PSO optimization method to tune the controller gain settings. In addition, a secondary PID controller was constructed for the proposed power system. The PSO outcomes in comparison to other prevalent approaches, such as the conventional, GA, and DE algorithm, was performed to prove its supremacy. The major advantage of PSO is that it can avoid premature convergence to local minima and also provide high-quality solutions. The main advantage of PSO is that it requires fewer parameters to tune. PSO obtains the best solution from particle interactions with a high-dimensional search space.

The PSO–PID controller was employed in this study with different cost functions. The system's performance was evaluated by comparing its response to one of a conventional-, genetic algorithm- and differential evolution algorithm-tuned PID controller for an identical power system. The primary goal and motivation for this study were to advance the performance of the proposed system and maintain system stability in critical situations to provide high-quality power to all consumers. To improve the system parameters, a PSO-tuned PID controller was used with four different cost functions to overcome the crisis. The PSO–PID response was also examined by comparing it to conventional-, GA- and DE-based PID controller responses under the same criterion.

*1.2. Main Contribution*

In this study an isolated power system was integrated with certain conventional sources of energy production. The current work was only designed and explored taking into account a single cost function. In contrast, this study closes this gap by analyzing the performance of the proposed controller and optimization approach in the addressed power system with four distinct cost functions. To determine the precise superiority of the suggested improved controller, a comprehensive investigation was conducted.

### 1.3. Main Highlights

- An isolated power system that includes the major and most common sources of electrical power was developed for investigation.
- A secondary controller (PID) was designed to implement the LFC in the proposed power system.
- The PSO technique was used to tune the controller gain parameters, to enhance its performance with the support of four different cost functions: IAE, ISE, ITAE and ITSE.
- The results were compared among the cost functions and conventional-, GA-, and DE-tuning method-based PID controller.

### 1.4. Article Organization

This article is organized as follows: Section 1 reviews the literature related to the current study and many optimization strategies for the used controller. The mathematical modelling of the studied power system and the associated Simulink model are presented in Section 2. In Section 3, details of the used controller, conventional-, GA-, DE- and PSO-tuning methods are given. In Section 4, the performance of the proposed solution is presented and discussed. In the Section 5, the main findings of the current study are presented.

## 2. Mathematical Modeling of the Power System

The proposed power network contains three varieties of power-generating units: thermal, hydro, and gas units. Additionally, all three units were considered as a source of a single system and a PID controller was introduced to regulate the oscillation. The Simulink model of the proposed system arrangement is presented in Figure 1. As discussed in [8,9], the mathematical expression of the proposed system is:

Thermal power system components:

$$\text{Governor} \ = \ \frac{1}{1 + sT_{sg}} \tag{1}$$

$$\text{Reheater} \ = \ \frac{1 + sK_rT_r}{1 + sT_r} \tag{2}$$

$$\text{Steam turbine} \ = \ \frac{1}{1 + sT_t} \tag{3}$$

Hydro power system components:

$$\text{Governor} \ = \ \frac{1}{s\,T_{gh} + 1} \tag{4}$$

$$\text{Drop compensation} \ = \ \frac{T_r + 1}{T_{rh}\,s + 1} \tag{5}$$

$$\text{Penstock turbine} \ = \ \frac{-T_w s + 1}{0.5\,T_w\,s + 1} \tag{6}$$

Gas power system components:

$$\text{Valve positioner} \ = \ \frac{1}{s\,B_g + C_g} \tag{7}$$

$$\text{Speed governor} \ = \ \frac{X_g\,s + 1}{Y_g\,s + 1} \tag{8}$$

$$\text{Combustion reaction} \ = \ \frac{-T_{cr}\,s + 1}{T_F\,s + 1} \tag{9}$$

$$\text{Compressor discharge} = \frac{1}{s\,T_{CD} + 1} \tag{10}$$

where $T_{sg}$, $T_r$, and $T_t$ represent the governor, reheater, and steam turbine time constants, respectively. $T_{gh}$, $T_r$, $T_{rh}$, and $T_w$ are the time constants of the governor, drop compensation, and penstock turbine, respectively in a hydro power plant. $B_g$ and $C_g$ are the gas turbine constant of the valve positioner and the gas turbine valve positioned, respectively. $X_g$ and $Y_g$ are the lead and lag time constants of the governor. $T_{cr}$ is the combustion reaction time delay, $T_f$ is the fuel time constant, and $T_{CD}$ denotes the compressor discharge volume time constant.

From the transfer functions of the designed power network, the Simulink model was developed for investigation by using the MATLAB/Simulink environment for frequency regulation. The investigation was conducted by applying a 1% step load perturbation (SLP). The controller gain value of the PID controller was optimized and implemented by writing a separate PSO technique-coding mfile. The nominal values for the system parameters were adopted from [8,9].

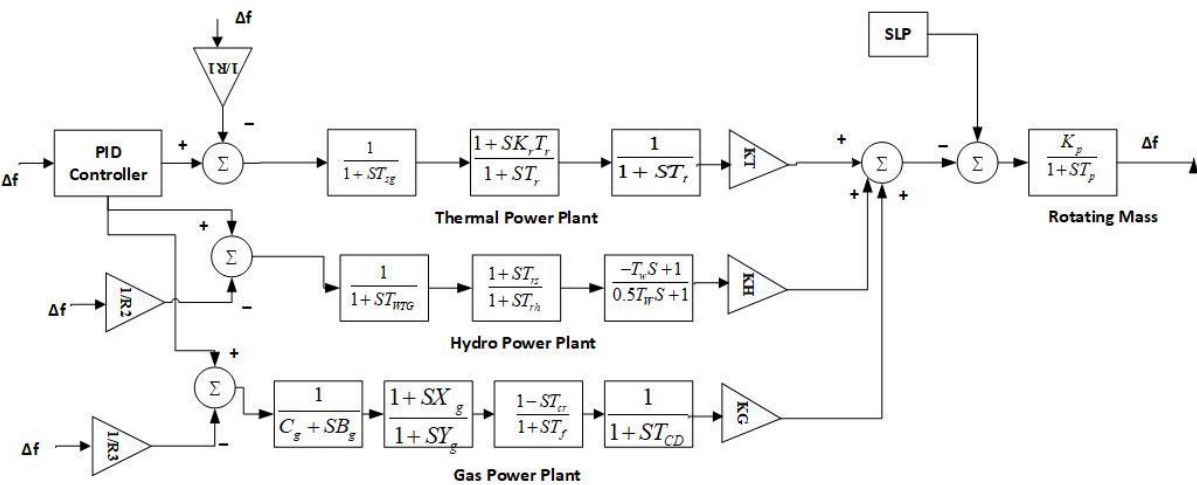

**Figure 1.** Mathematical model of the proposed system.

## 3. Control Strategy

The PID controller is a very common controller in the field of control and automation. The design, implementation, and handling are easy in the PID controller. It has the ability to self-tune for the determined value. According to [3,4], the mathematical function of the PID controller is:

$$G(s) = K_p + \frac{K_i}{s} + sK_d \tag{11}$$

where $K_p$, $K_i$, and $K_d$ are the gain of proportional, integral controller, and derivative controller, respectively.

### 3.1. Cost Functions

The cost functions are the scale that helps to optimize the controller gain parameters, which support the used the optimization technique to obtain the optimized values and improve system performance. Based on the error function the controller controls the oscillation. Here, four cost functions: IAE, ISE, ITAE, and ITSE, were used to find the optimal control gain values.

According to [4], the mathematical expression for each of the used cost functions is:

$$J_{IAE} = \int |ACE|\, dt \tag{12}$$

$$J_{ITAE} = \int t.\,|ACE|\, dt \tag{13}$$

$$J_{ISE} = \int \{ACE\}^{^2} dt \tag{14}$$

$$J_{ITSE} = \int t. \{ACE\}^{^2} dt \tag{15}$$

where ACE is the area control error and t is the simulation time.

In this work, the PID controller gain parameters were obtained based on the adopted cost functions and using the conventional, genetic algorithm, differential evolution algorithm, and PSO technique for analyzing the performance of the LFC of the proposed system. The conventional-tuning method and PSO techniques are discussed in the following section.

### 3.2. Methodology
#### 3.2.1. Conventional Tuning Method

To obtain the optimal controller gain value, the conventional method of trial and error-tuning was used. The first integral gain value ($K_I$) was tuned using the trial and error procedure in this tuning process. After determining the optimal gain value for $K_I$, it was set as a constant value. Following this, the proportional gain value ($K_P$) was tuned to achieve its optimal value, just like the integral gain value. Following this, the $K_I$ and $K_P$ were fixed as constants, and the derivative controller gain value ($K_D$) was adjusted as suggested by [3,4]. Figures 2–5 depict the curves as to the performance indices for the IAE, ISE, ITAE, and ITSE, respectively.

The optimal values of the PID controller gain were obtained at the end of the tuning process using the various studied cost functions, shown in Table 2.

**Table 2.** Gain values of the conventional PID controller.

| Objective Function/Optimized Gain | $K_p$ | $K_i$ | $K_d$ |
|:---:|:---:|:---:|:---:|
| IAE | 1.1 | 0.4 | 0.003 |
| ISE | 1.2 | 0.4 | 0.11 |
| ITAE | 0.6 | 0.364 | 0.1 |
| ITSE | 1 | 0.4 | 0.03 |

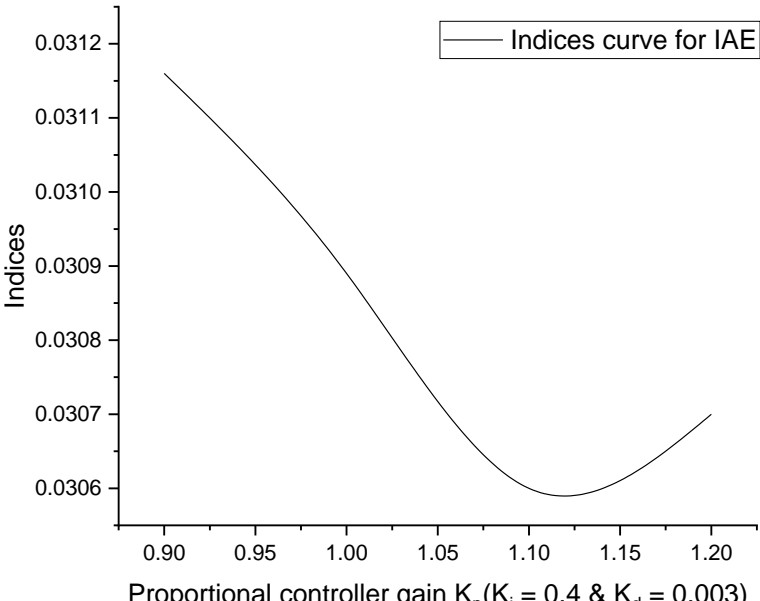

**Figure 2.** IAE performance indices curve.

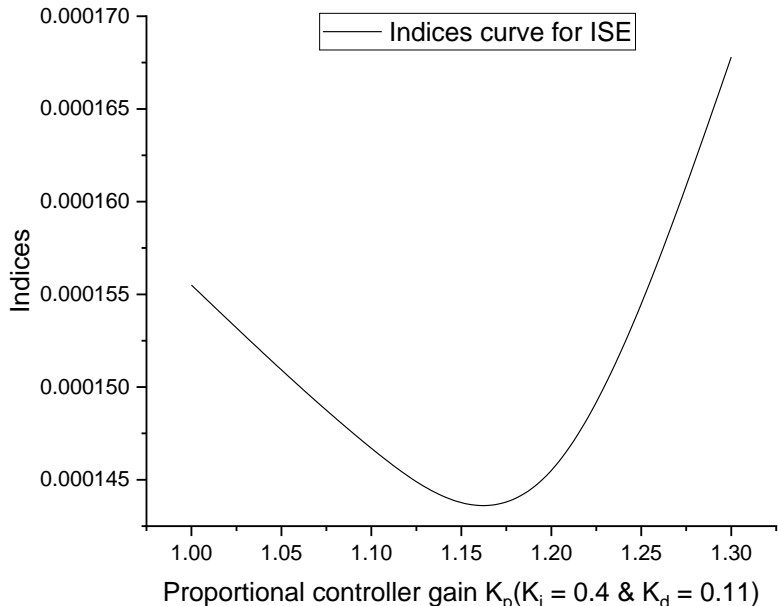

**Figure 3.** ISE performance indices curve.

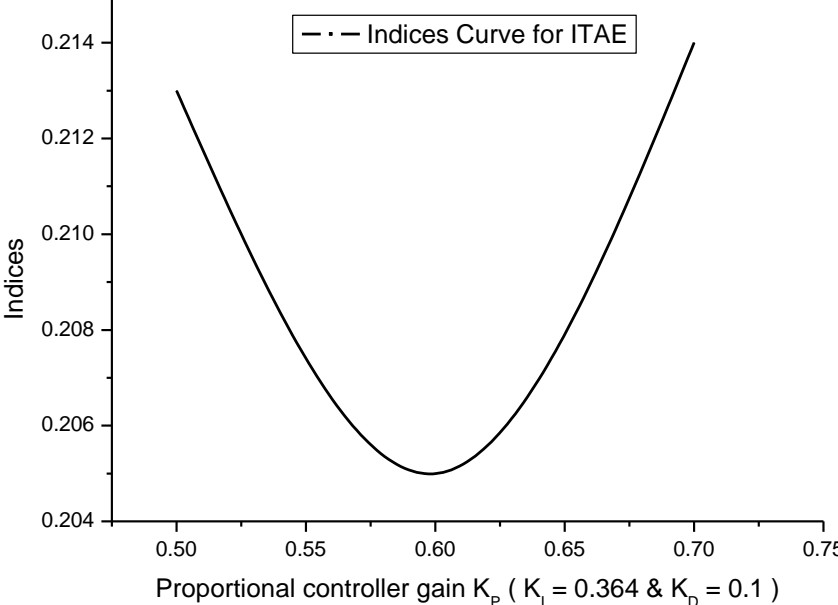

**Figure 4.** ITAE performance indices curve.

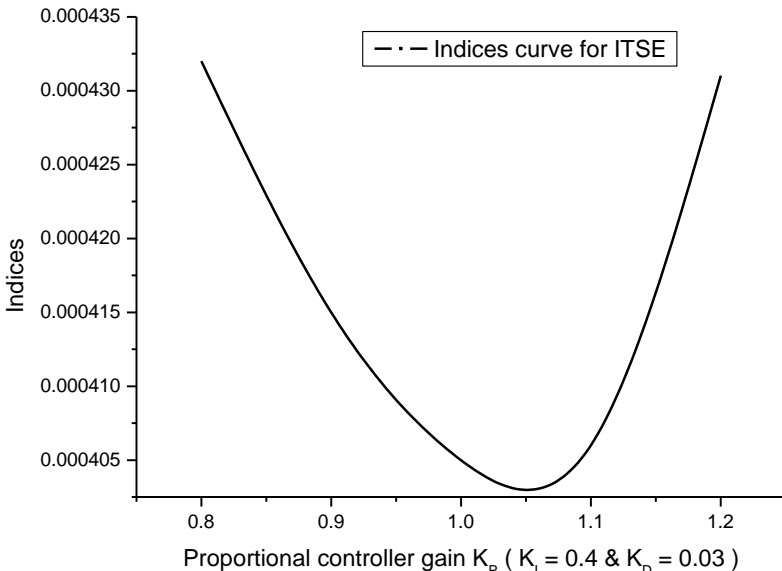

**Figure 5.** ITSE performance indices curve.

### 3.2.2. Particle Swarm Optimization Tuning Method

The PSO algorithm was proposed in 1995 by Dr. Kennedy and Dr. Eberhart based on the community behavior of the bird flocking/fish schooling process. In the PSO approach, each solution is called a 'particle' in the search area. The tuning process depends on the fitness values of the objective function. The particle's momentum directs the particles' path through the space they are searching for the best individual and global values. A collection of random particles initializes the swarm and then looks for optimization by changing iterations. The personal and global best concepts are applied to every particle to achieve optimal gain values. Every iteration has achieved the best solution called the local best. Each local best value is optimized and the final best value obtained is called the global best [3,10]. The flow chart of the algorithm used for the PID controller tuning is shown in Figure 6 [26].

The appropriate gain values for the PID controller were determined after the tuning phase to maintain power system stability amid unforeseen load changes or emergency scenario settings [41]. Table 3 presents the gain values obtained from PSO tuning.

**Table 3.** Gain values of the PSO-optimized PID controller.

| Objective Function/Gain Value | $K_p$ | $K_i$ | $K_d$ |
|:---:|:---:|:---:|:---:|
| IAE | 0.9934 | 0.9997 | 0.0915 |
| ISE | 0.9967 | 0.9997 | 0.1315 |
| ITAE | 0.7741 | 0.9994 | 0.1850 |
| ITSE | 0.9769 | 0.9999 | 0.0843 |

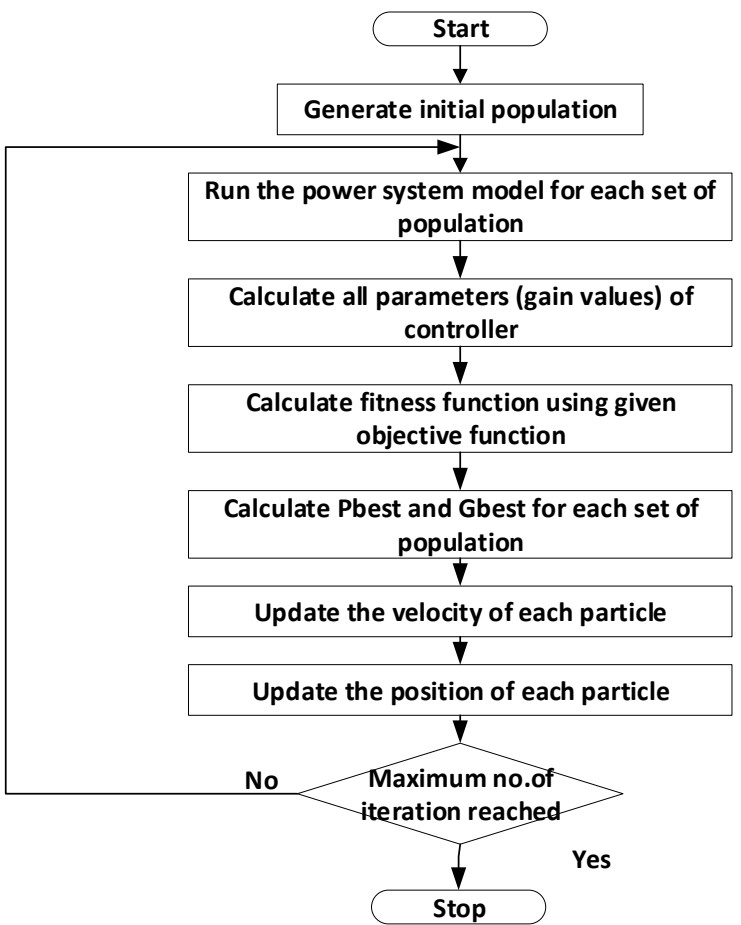

**Figure 6.** Flow chart of the PSO for PID controller tuning.

## 4. Simulation Result

Using MATLAB 2016a, the simulation model of the proposed system was developed. This section analyzes the performances of the conventional, GA, DE, and PSO–PID controller replies. Using 1% SLP, the performance of the system was evaluated for different optimization techniques. Finally, the PSO response was assessed by comparing it to the response of a PID controller based on a conventional-, GA-, and DE-tuning approach. The frequency response of the comparison with various cost functions is shown in Figures 7–10. Red, green, brown and black lines represent the conventional-, GA-, DE- and PSO-tuned PID controller responses, respectively. The corresponding controlled numerical values to the frequency deviation (delF) are presented in Tables 4–7, where Ts represents the settling time, POS represents the peak over shoot and PUS represents the peak under shoot.

**Table 4.** Controlled parameters of delF using the ISE cost function.

| Optimization Methods/Controlled Parameters | Ts (Seconds) | POS (Hz) | PUS (Hz) |
|:---:|:---:|:---:|:---:|
| Conventional | 88 | $0.2 \times 10^{-3}$ | $6.6 \times 10^{-3}$ |
| GA | 46 | $0.1 \times 10^{-3}$ | $6.5 \times 10^{-3}$ |
| DE | 52 | $0.2 \times 10^{-3}$ | $6.5 \times 10^{-3}$ |
| PSO | 45 | $0.1 \times 10^{-3}$ | $6.4 \times 10^{-3}$ |

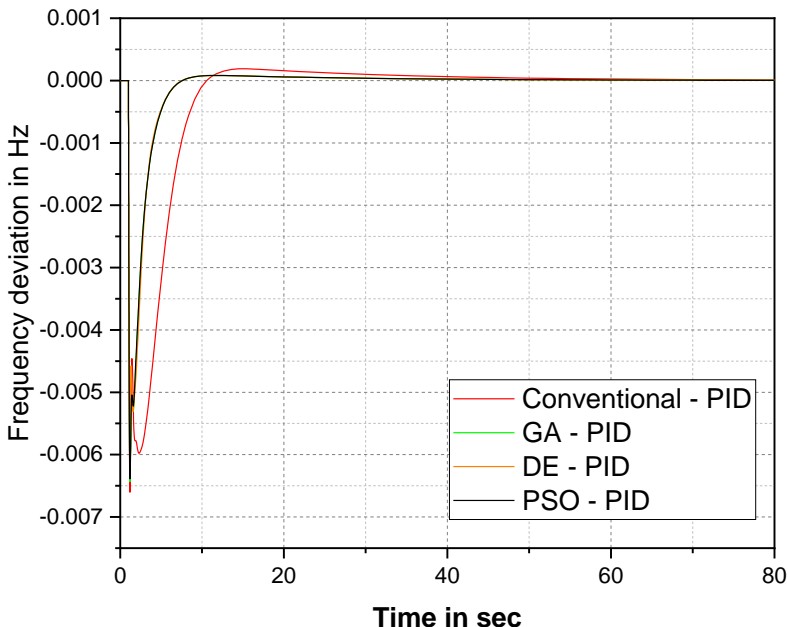

**Figure 7.** delF comparison for the ISE cost function.

The response of the frequency deviation for different cost functions was analyzed. From the detailed analysis, the PSO–PID controller with the ISE cost function provided better results than the other techniques. The PSO–PID with ISE settled the oscillation at 45 s, quicker than the others. Improvement of the proposed controller over the conventional was 104%, over the GA was 7%, and over the DE was 21%. The Figure 8 shows that delF comparison for IAE cost function and the numerical values from Figure 8 is reported in Table 5.

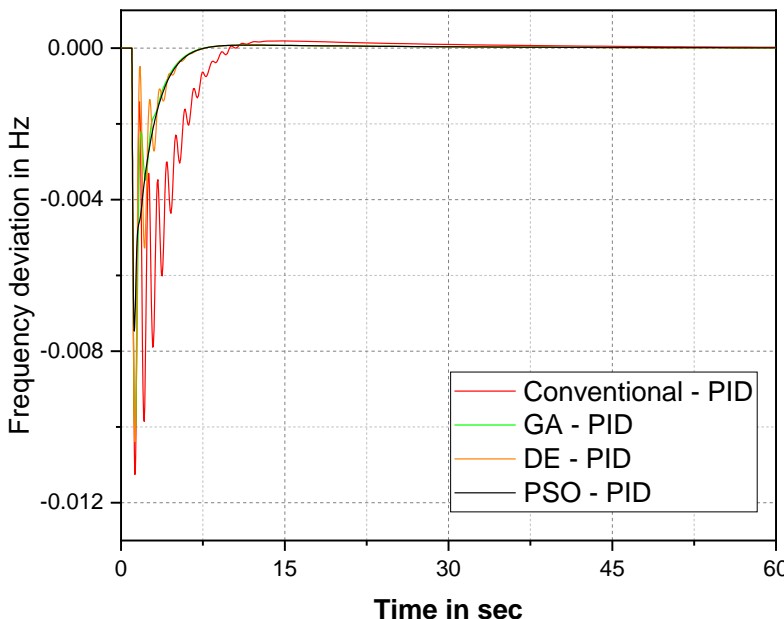

**Figure 8.** delF comparison for the IAE cost function.

**Table 5.** Controlled parameters of delF using the IAE cost function.

| Optimization Methods/Controlled Parameters | Ts (Seconds) | POS (Hz) | PUS (Hz) |
|:---:|:---:|:---:|:---:|
| Conventional | 58 | $0.2 \times 10^{-3}$ | $11.2 \times 10^{-3}$ |
| GA | 50 | $0.1 \times 10^{-3}$ | $10.5 \times 10^{-3}$ |
| DE | 42 | $0.1 \times 10^{-3}$ | $10.5 \times 10^{-3}$ |
| PSO | 40 | $0.1 \times 10^{-3}$ | $7.5 \times 10^{-3}$ |

The graphical and numerical response comparison of the proposed controller with the IAE cost function in Figure 8 and Table 5, from this PSO–PID controller performance is dominant over the other optimization techniques. It controlled the oscillation for 40 s. In terms of percentage improvement, the PSO–PID controller was 45% better over the conventional, 25% over the GA, and 5% over the DE. Figure 9, represents delF comparison for ITSE cost function. The numerical values from the Figure 9 is reported in Table 6.

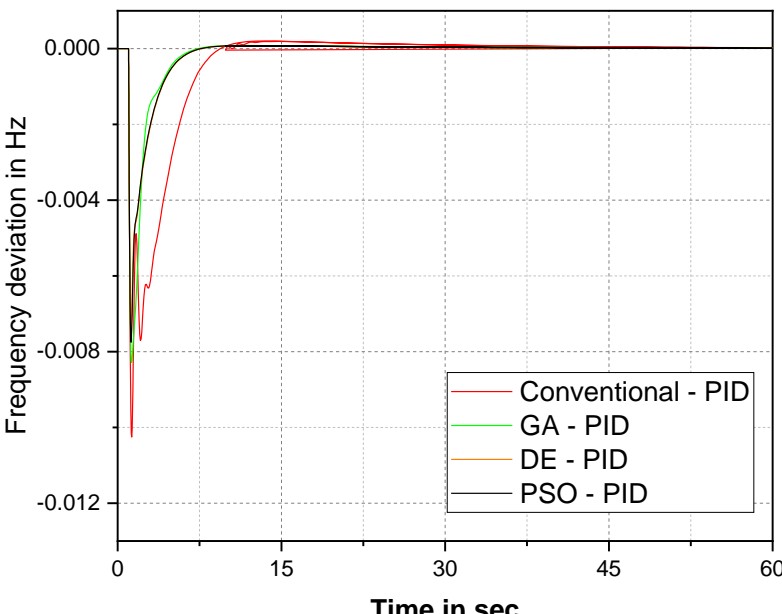

**Figure 9.** delF comparison for the ITSE cost function.

**Table 6.** Controlled parameters of delF using the ITSE cost function.

| Optimization Methods/Controlled Parameters | Ts (Seconds) | POS (Hz) | PUS (Hz) |
|:---:|:---:|:---:|:---:|
| Conventional | 58 | $0.2 \times 10^{-3}$ | $10.2 \times 10^{-3}$ |
| GA | 45 | $0.1 \times 10^{-3}$ | $8.4 \times 10^{-3}$ |
| DE | 42 | $0.1 \times 10^{-3}$ | $7.5 \times 10^{-3}$ |
| PSO | 39 | $0.1 \times 10^{-3}$ | $7.7 \times 10^{-3}$ |

The ISTE cost function-based frequency response comparison in Figure 9 and Table 6 demonstrate that the PSO–PID controller provides better results than the other optimization techniques. The PSO–PID controller settled the oscillation at 39 s. The improvement of the PSO–PID controller over the conventional method was 49%, over GA was 15, and over DE was 8%. The graphical and numerical comparison of del F for ITAE cost function is given in Figure 10 and Table 7 respectively.

**Table 7.** Controlled parameters of delF using the ITAE cost function.

| Optimization Methods/Controlled Parameters | Ts (Seconds) | POS (Hz) | PUS (Hz) |
|---|---|---|---|
| Conventional | 59 | $0.2 \times 10^{-3}$ | $14 \times 10^{-3}$ |
| GA | 51 | $0.1 \times 10^{-3}$ | $9.3 \times 10^{-3}$ |
| DE | 41 | $0.05 \times 10^{-3}$ | $10.6 \times 10^{-3}$ |
| PSO | 33 | $0.05 \times 10^{-3}$ | $6.7 \times 10^{-3}$ |

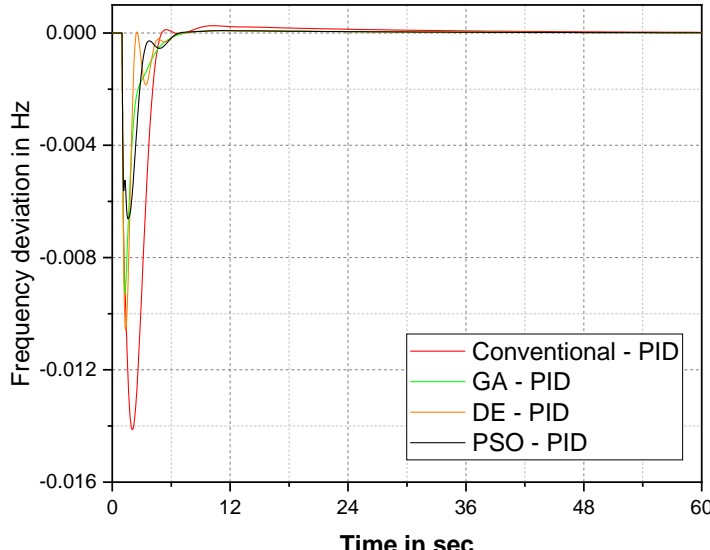

**Figure 10.** delF comparison for the ITAE cost function.

The performance comparison of the PSO–PID controller with other optimization techniques in Figure 10 and Table 7, provides better results over the conventional, GA, and DE algorithms. The ITAE-based PSO–PID controller settled the oscillation at 33 s. The percentage improvement of the PSO–PID controller over the conventional method was 79%, over GA was 55%, and over DE was 24%. A bar chart comparison between the four cost functions settling time shown in Figure 11.

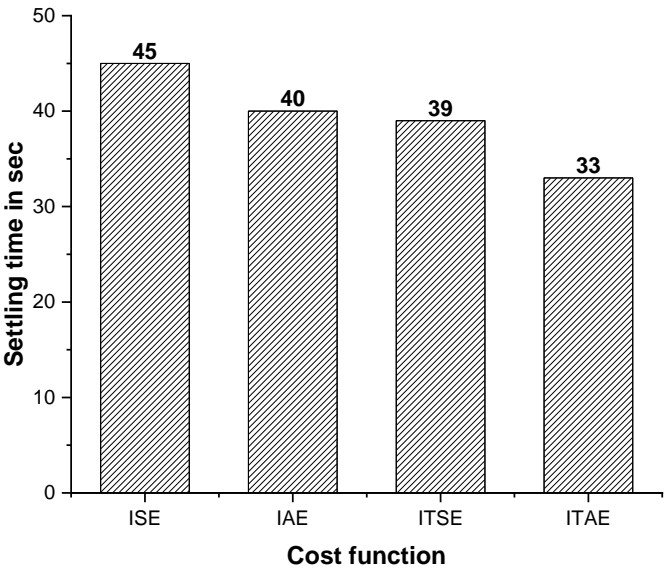

**Figure 11.** Bar chart comparison of settling time (PSO–PID controller).

Finally, the PSO–PID controller response with all four cost functions was compared by a bar chart in Figure 11. It shows that the ITAE cost function-based PSO–PID controller provides a better response than other cost function by means of fast settled response over other cost functions (IAE, ISE and ITSE).

## 5. Conclusions

A complete performance study of the LFC for a single-area, multi-source power-generating unit with a secondary PID controller was presented in this work. Gain settings for PID controllers were estimated using traditional tuning, GA, and DE algorithms as well as the PSO technique with four distinct cost functions. An evaluation of the system response comparisons demonstrated that a PSO–PID controller with an ITAE cost function-based controller generates a better-regulated response than the conventional-, GA-, or DE algorithm-tuned controller response. Similarly, the response of the PSO–PID controller with different cost functions was compared. The PSO–PID controller with the ITAE cost function was more dominant than other cost functions and optimization technique controller responses.

**Author Contributions:** B.D.; Conceptualization, writing—original draft preparation, J.K.; writing—review and editing; A.B.; validation and investigation, N.D.; formal analysis and supervision, J.M.R.S.T.; supervision and resources. All authors have read and agreed to the published version of the manuscript.

**Funding:** This research received no external funding.

**Institutional Review Board Statement:** Not applicable.

**Informed Consent Statement:** Not applicable.

**Data Availability Statement:** Not applicable.

**Conflicts of Interest:** The authors declare no conflict of interest.

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
