# Peer review of "Load Frequency Control Assessment of a PSO-PID Controller for a Standalone Multi-Source Power System"

_technologies, doi:10.3390/technologies11010022_

Round 1

Reviewer 1 Report (Previous Reviewer 1)

The work is well written, good additions. The lack of a laboratory experiment is its shortcoming.

Author Response

Thanks for your encouraging comments.

Reviewer 2 Report (New Reviewer)

I deeply analyzed previous reviews and responses. Based on my assessment the reviewers suggest adequate changes. The authors correctly revised the paper.

My recommendation is to accept this paper after minor revision.

The only minor revision is to include the main results in the abstract in a quantitative approach. 

Author Response

Thanks for your encouraging comments and acceptance

Reviewer 3 Report (New Reviewer)

The revised paper is fine. I suggest acceptance of this paper. 

Author Response

Thanks for your encouraging comments and acceptance

This manuscript is a resubmission of an earlier submission. The following is a list of the peer review reports and author responses from that submission.

Round 1

Reviewer 1 Report

Missing Lab Experiment or "Micro Grid". Such results clearly confirm the usefulness of the method. It is worth showing the digital version of the analyzed algorithms. Only these can be used.

Reviewer 2 Report

This paper addresses a topic related to load-frequency control of power systems with multiple generation sources. The objective is to apply PSO to adjust the common PID controller to 3 generation units: Hydraulic, gas and thermal. Based on the work presented, some points can be emphasized:

1- The authors carry out a bibliographic review having the majority of citations from 2020. Only one citation of an article from 2022 in conference. Therefore, citations from 2021 and 2022 in journals and/or conferences would be interesting. Based on this review, the present work presents a marginal contribution with the application of the PSO in a PID adjustment problem.

2- The proposed methodology could have been better presented, showing an equation for Particle Swarm optimization. What is the advantage over other methods?

3- This reviewer considers that the comparison of results presented is insufficient. The results of the proposed method (PSO) should have been compared with other meta-heuristics found in the literature. What is the advantage of the proposed method in relation to already exists in the Literature? What's the advance? Comparisons were made only in relation to the so-called “conventional method” to define PID gains.

4- The authors use the term “Area Control Error” (ACE) in their work, including graphs showing the ACE obtained in the simulations. However, as the system used is a system with a single control area, the ACE is simply the frequency deviation of the system, since there is no interchange. Therefore, it was not clear what the authors were referring to when mentioning the ACE, since they presented different values for the ACE and for the frequency deviation (figures 10 and 11).

5- The definition of a single PID for 3 units of different types seems only theoretical considering that these units are far from each other. Furthermore, the valve opening/closing speed constraints of these turbines are different. So they cannot have the same PID controller.

6- Authors should pay attention to some grammatical and organization errors in the article. Examples:

• equation 3 “stream turbine” should be “steam turbine”. The same error occurs elsewhere in the text;

• Description of article organization at the end of section 1 is wrong: confusion in section numbers and their respective contents;

• Acronyms not defined throughout the work;

• Variables typed incorrectly;

• Confusing text in some sections

Based on the work presented, this reviewer considers that this paper does not present a contribution to be published.